# Microstructure arrays of DNA using topographic control

Yun Jeong Cha[1,4], Soon Mo Park[1,4], Ra You[1], Hyoungsoo Kim[2] & Dong Ki Yoon [1,3]

DNA is a common biomaterial in nature as well as a good building block for producing useful structures, due to its fine feature size and liquid crystalline phase. Here, we demonstrate that a combination of shear-induced flow and microposts can be used to create various kinds of interesting microstructure DNA arrays. Our facile method provides a platform for forming multi-scale hierarchical orientations of soft- and biomaterials, using a process of simple shearing and controlled evaporation on a patterned substrate. This approach enables potential patterning applications using DNA or other anisotropic biomaterials based on their unique structural characteristics.

[1] Graduate School of Nanoscience and Technology, Korea Advanced Institute of Science and Technology, Daejeon 34141, Republic of Korea. [2] Department of Mechanical Engineering, Korea Advanced Institute of Science and Technology (KAIST), Daejeon 34141, Republic of Korea. [3] Department of Chemistry and KINC, Korea Advanced Institute of Science and Technology, Daejeon 34141, Republic of Korea. [4] These authors contributed equally: Yun Jeong Cha and Soon Mo Park. Correspondence and requests for materials should be addressed to D.K.Y. (email: nandk@kaist.ac.kr)

DNA is produced in enormous quantities in plants[1] and animals[2,3] every year, and as a result DNA material extracted from nature is very cheap compared with general polymer materials. However, the quality of such DNA is not good enough for use in bioscience[4] or medical engineering[5,6]. But although the lengths of the DNA chains and the nucleotide sequences of crude DNA material are varied, the structural features of DNA, such as chirality, periodicity and diameter, are the same as well-prepared or expensive synthetic DNA materials. Indeed, the chemical and topographic characteristics of DNA material, like the negatively charged backbone of the DNA double helix, enable such DNA materials to be used in the fabrication of complex structures when combined with versatile guest functional materials such as biomaterials[7,8], particles[9–11] and liquid crystals (LCs)[12–14]. But for this purpose, depending on the specific application, well-ordered and oriented DNA chains are required.

The ordering and orientation of DNA is based on the LC characteristics of the DNA material, which is related to the concentration of the aqueous DNA solution. For example, when it is high enough to form a LC phase, ~50 mg/ml, it shows a nematic (N) phase[15,16]. As the concentration increases up to ~300 mg/ml, the DNA solution adopts a columnar (Col) phase[15,17]. When the concentration is further increased by evaporation of the solvent, such as water, a crystallised columnar phase (Cr) is formed, with little structural change[17]. Several methods have been proposed for preparing well-ordered and oriented DNA material in the LC phase, including simple drying[18,19], mechanical shearing[20] or combing[21,22], topographic confinement[23] and magnetic ordering[24]. However, these techniques only form relatively simple structures, such as unidirectional aligned DNA textures or irregular zigzag morphologies.

Here, we fabricated periodic arrays of various microstructures of crude DNA material, using the doctor blade method on a microposts-patterned silicon (Si) substrate. By spatially controlling water evaporation, the isotropic (Iso) to Cr phase transition of the DNA was directly manipulated to form various kinds of interesting textures. The resultant aligned Col LC or Cr phase of DNA showed connected knit-like morphologies, with arrays of regular 'ice cream cone'-like patches, and tilted comet patterns.

## Result

### Generation of DNA microstructures on microposts-substrates.
The DNA material used in this study was salmon sperm DNA duplex (purchased from Sigma Aldrich, St. Louis, MO, USA) with a Gaussian distribution in base pairs, whose centre point was ~2000 base pairs with a corresponding contour length of ~680 nm[25]. This DNA is regarded to be a semi-flexible polymer with a persistence length of ~ 50 nm[15].

When a droplet of aqueous DNA solution (25 mg/ml) was spread on a flat silicon substrate, the DNA chains aligned with the contact line, which minimises elastic energy during water evaporation (Fig. 1a, b)[19,20,26]. The director of the DNA chain ($n_{DNA}$) was directly determined by polarised optical microscopy (POM) using a first-order retardation plate ($\lambda = 530$ nm) at 45° between the cross polarisers (Fig. 1b–e). Magenta-coloured domains appeared when the $n_{DNA}$ was isotropic or aligned parallel with either of the polarisers. Blue or yellow domains in the POM images mean that the $n_{DNA}$ is oriented perpendicular or parallel to the slow axis (pink arrows in the insets of the POM images) of the retardation plate, respectively[20]. The enlarged red, blue and green solid line-boxed domains are cropped to show how the DNA chains are aligned (Fig. 1c–e).

Unlike this simple and trivial arrangement of DNA, when the same aqueous DNA solution was dropped on a microposts-

patterned silicon substrate (Fig. 1f), which produced a geometric confinement effect, similar but totally different coloured domains formed over a large area, ~7 mm² as the water evaporated (Fig. 1g–j). This was quite different than the results in a previous study on the lyotropic LC phase performed in the steady state condition[27]. The DNA molecules on the posts substrate followed the capillary flow as the liquid component evaporated. Finally, the DNA molecules aligned along the contact line of the solution, which is consistent with the previous results (Fig. 1b). In the enlarged images (Fig. 1h–j), however, it can be seen that completely different structures have been generated depending on the dried angle, exhibiting knit-like and ice cream cone-like morphologies.

### Topographic control using microposts and templating angle.
To investigate each of the textures generated in Fig. 1h–j, we carried out controlled experiments by varying the pulling direction (PD) of the upper glass substrate with a fixed bottom post-patterned substrate (separation, $s = 10$ μm, diameter, $D = 5$ μm, height, $h = 5$ μm) (Fig. 2a). An angle ($\varphi$) was defined between the PD and the parallel aligned post direction ($n_{post}$), and three angles, of 0°, 45° and 67.5° were chosen to generate representative micro-textures (Fig. 2b–k). In the beginning of this work, various pulling speeds were tested, and it was found that 3–7 μm/s made the reliable microstructures, whereas slower speeds did not form regular structures. The fastest speed made interesting structures but did not generate reliable or microposts-dependent morphologies (Supplementary Fig. 1). Therefore, the pulling speed was fixed at 3 μm/s in this study.

A schematic sketch of each case is provided with the corresponding colours observed in the POM experiments to show the $n_{DNA}$ (Fig. 2b): the black dotted line indicates the $n_{DNA}$. At $\varphi = 0°$, unidirectional knit-like structures were generated at the Iso-N-Col phase transition as the water evaporated (Fig. 2c–e). In the enlarged inset images, the formation of blue and yellow domains can be clearly observed, indicating that the DNA chains are aligned to form a V shape between the neighbouring posts, as described in the first row in Fig. 2b, c[21]. The sequential growth of each pattern is recorded in Supplementary Movie 1.

The underlying mechanism of this fabrication process can be understood by observing contact lines where the phase transition happens to generate the LC and Cr phases. This was accomplished using reflection mode optical microscopy without polarisers, as shown in Fig. 3a–h. The corresponding contact lines are schematically described in Fig. 3i, j. The aqueous DNA solution meniscus formed by the microposts and upper slide glass moves to generate the three kinds of domains. The relatively dark domains result from the scattering effect of the hydrated film under the height of the posts, and reveal moving contact lines, which evaporate to form the aligned LC phases. What should be noted here is that the lines are not straight but curved, which is important for the sequential growth of the array of DNA microstructures with the V shape at $\varphi = 0°$ (Fig. 2c–e, Supplementary Movie 2). Similar behaviour at $\varphi = 45°$ can be found, and the domains in Fig. 3j are important for understanding the sequential changes in Fig. 2b, f–h. The detailed meniscus at the α and β areas can be seen in fluorescent confocal polarising microscopy images (Fig. 3k and l).

The dark lines clearly shown in the yellow boxes in Fig. 2c–e look like domain wall defects where the propagation direction of the $n_{DNA}$ is changed, as previously reported[19,21]. However, by rotating the sample with either of the polarisers (Supplementary Fig. 2 and Supplementary Movie 3) we realise the black lines are not defect walls, but parallel aligned DNA chains. The V shaped texture is also changed to an 'ice cream cone'-like morphology

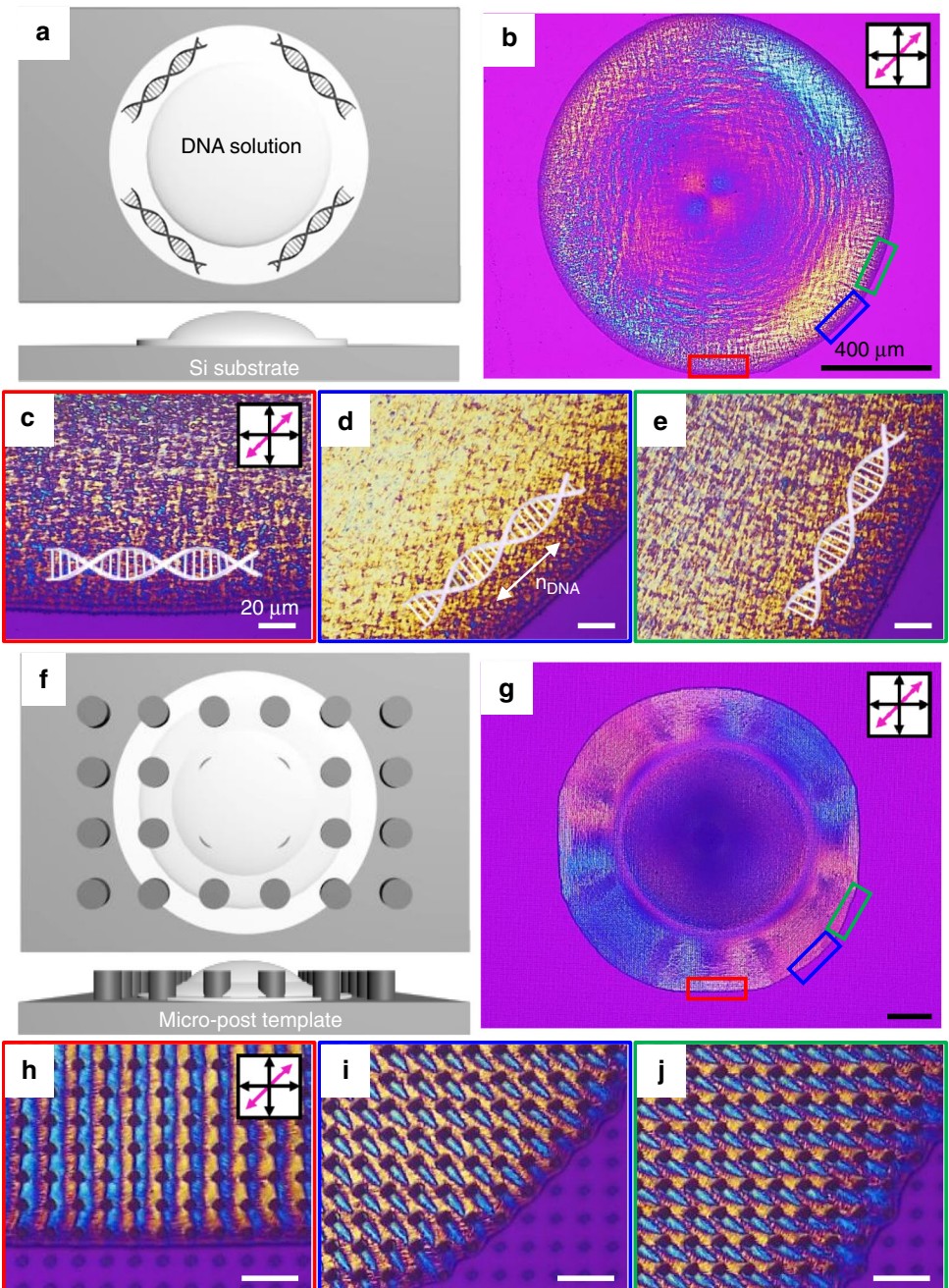

**Fig. 1** Fabrication scheme and POM images of DNA films on a flat/micropost-patterned substrate. **a** Simply dried DNA film on the flat Si substrate and **b** its POM image. **c–e** Enlarged images from **b** show distinctive colours depending on the position of the DNA film, and the corresponding orientation of DNA chains. **f** Hydrated DNA film on the microposts and **g** its POM image. **h–j** Enlarged images from **g** show the different patterned textures depending on the drying direction. Black scale bars represent 400 μm and white ones are 20 μm

when the sample is rotated at 45°, although the blue domains are connected to neighbouring posts along with the PD (Supplementary Fig. 2c). In more detail, there are two kinds of black lines in Fig. 2c–e; the first is from the connection between neighbouring posts along with PD, which is very clear to see (α in the inset of Fig. 2e). The second type is generated in between the microposts (β in the inset of Fig. 2e), which appears relatively hazy compared with the first, α lines. The first case is the product of the micropost, which serves as a point to generate a wave or curves of DNA chains (Fig. 2b). This can be simply observed in the sequentially moving contact line (Fig. 3i, j), where the DNA chains follow the capillary force-induced flow to move toward the contact line. However, the orientation of the DNA chains is different than the movement of the contact line[20]. Indeed, the continuity of the contact line movement passing through the post and bottom substrate is quite different owing to the pinning effect resulting from the post geometry[28] (Supplementary Movie 2).

In contrast, the second, β domain results from the curved contact line where the Iso-LC transition happens, so that smoothly modulated DNA chains are arranged (Fig. 2b). The interesting area is γ (in the inset of Fig. 2d), where the isotropic region still remains even after the meniscus line passes. This can be understood using a schematic sketch (γ in Fig. 3i), where a temporary reservoir forms owing to the confinement of the microposts.

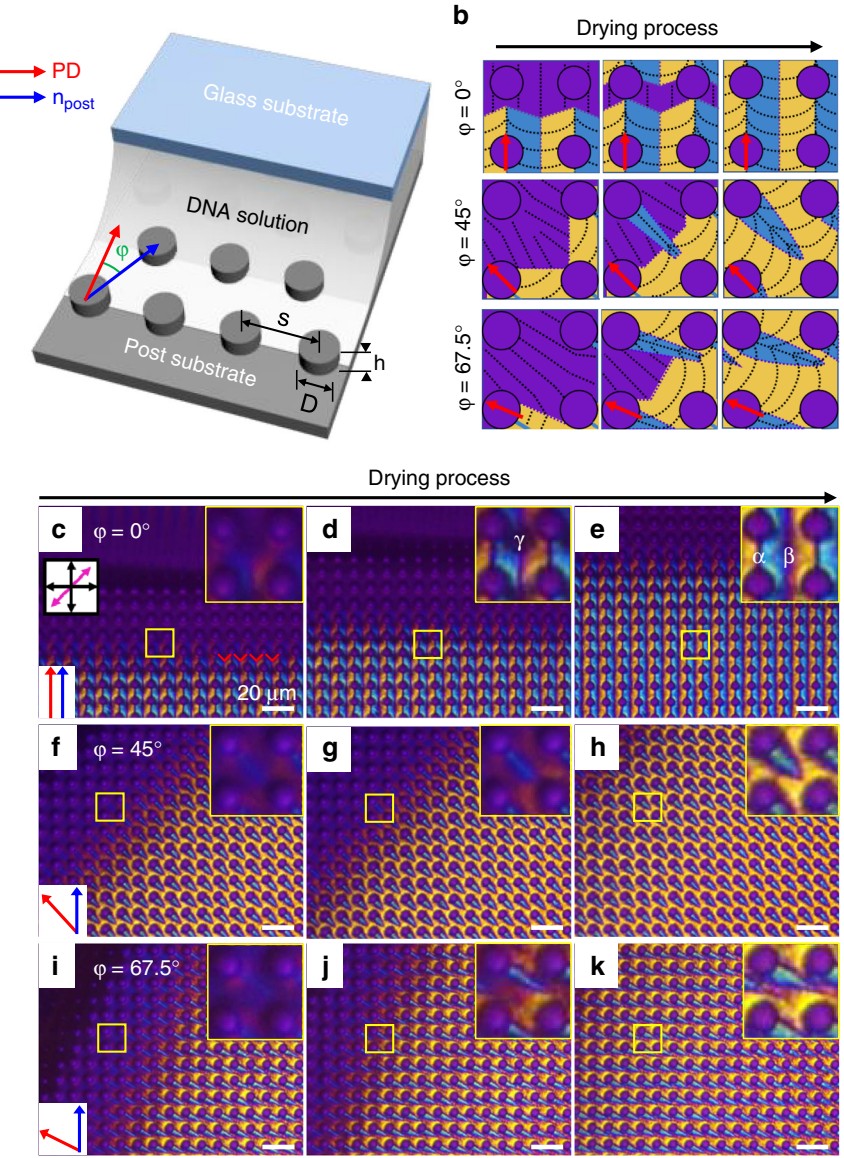

**Fig. 2 Real-time observation of fabrication process depending on the angle. a** Schematic illustration of how the experiments were performed using the shearing process, and the dimensions of the microposts ($s = 10$ μm, $D = 5$ μm and $h = 5$ μm). **b** Schematic sketches of the sequential changes in DNA chain-arrangement during the evaporation of water. **c–e** At $\varphi = 0°$, sequential growing microstructures of DNA film can be observed as the water evaporates. Similar but different structures can be generated **f–h** at $\varphi = 45°$, and **i–k** at $\varphi = 67.5°$. Enlarged images show the detailed orientation of the DNA chains. Yellow colour means the DNA chains are aligned parallel to the optical axis of the full wave plate (pink arrow in **c**), whereas the blue domains appear when the $n_{DNA}$ is perpendicular to the pink arrow. All scales are 10 μm

At $\varphi = 45°$, the ice cream cone-like structures are formed. The ice cream part indicates a single post, revealed by the magenta colour, and the cone part is a blue domain with a yellow background region, indicating a parallel aligned domain in the vertically aligned background of DNA chains with PD (Fig. 2f–h). The basic principle is just the same as the $\varphi = 0°$'s case, which is the reason the 45°-rotated sample shows a deformed V texture (Supplementary Fig. 2f).

More details can be seen in the schematic sketch (Fig. 3j), in which the sequential changes in the grey domains are repeatedly separated and merged as the contact line moves. During this process, the micropost acts as a hurdle to the contact line and does not allow DNA chains to align parallel with the contact line. The DNA chains are oriented perpendicular to the contact line behind the post. At $\varphi = 67.5°$, a behaviour similar to $\varphi = 0°$

can be found, and in the 45°'s case, where tilted comet-like blue domains are generated along the PD.

**Various kinds of microstructures by varying micropost-separation**. To extend the morphogenesis-versatility of our platform, the separation, s was varied, whereas $\varphi$ was fixed at 0°, and four kinds of separations were selected ($s = 7$, 10, 14 and 18 μm in Fig. 4). To determine $n_{DNA}$ clearly, the optical texture of the DNA material on each topographic pattern was observed (Fig. 4c–n) and the corresponding arrangement of DNA chains was sketched with blue, yellow and magenta colours (Fig. 4b). As s increased, the magenta areas denoted α and β in Figs. 2e and 4g became wider. This can be understood by the smoother curve of the DNA chains in the β area, which are aligned perpendicular to PD as s increases (Figs. 3g and 4b). For example, the β area at s =

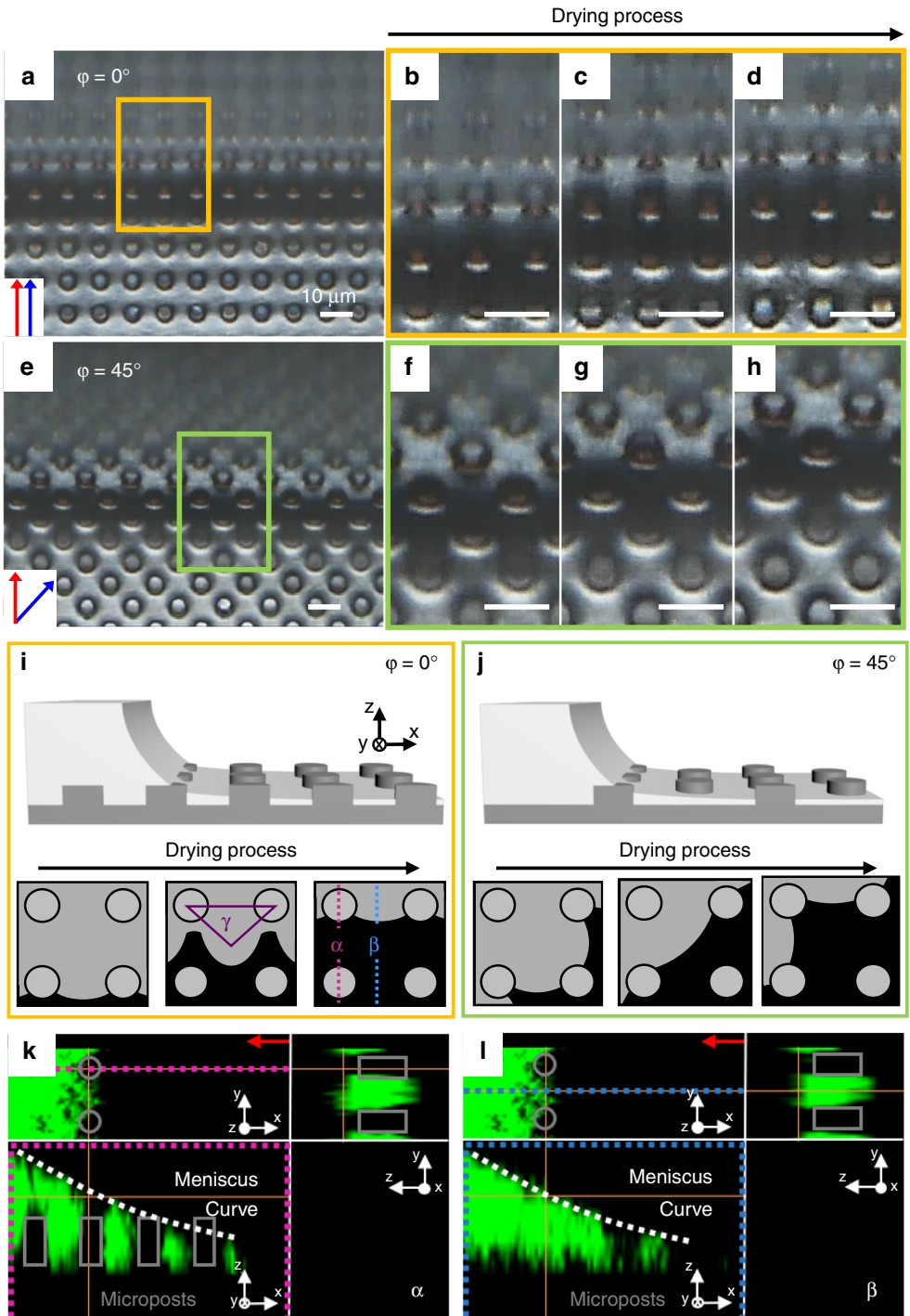

**Fig. 3** The in situ contact line changes during the process. Darker images can be observed when the meniscus generates at **a** $\varphi = 0°$ and **e** $\varphi = 45°$ and enlarged images in the colour boxes show the detailed contact line behaviour **b–d** $\varphi = 0°$ and **f–h** $\varphi = 45°$. The red arrow indicates the pulling direction. The drying process follows this direction, producing distinctive contact lines with varying $\varphi$. Schematic sketches as the DNA solution dries show how the contact lines were changed at **i** $\varphi = 0°$ and **j** $\varphi = 45°$. **k, l** Fluorescent confocal microscopy images show the cross section view at the α and β regions. The grey blocks indicate microposts. All scales are 10 μm

14 μm (Fig. 4i–k) is wider than that at $s = 7$ μm (Fig. 4c–e) because the DNA chains are aligned parallel to the flatter contact line. In the α area, fewer obstacles exist when s increases, which causes the extended alignment of DNA chains though the deviation to show a wider comet pattern with larger s. However, the DNA chains form zigzag patterns at $s = 18$ μm, which is quite different from the other results, and

this might result from undulating DNA chains, as shown in previous studies.

The A.G. Yodh group and O.D. Lavrentovich group reported similar undulating morphologies using sunset yellow[26] (SSY) and λ−DNA[19]. And we also showed the formation of zigzags of salmon sperm DNA chains over a large area using a simple brushing method[21]. In detail, these zigzags were the result of the

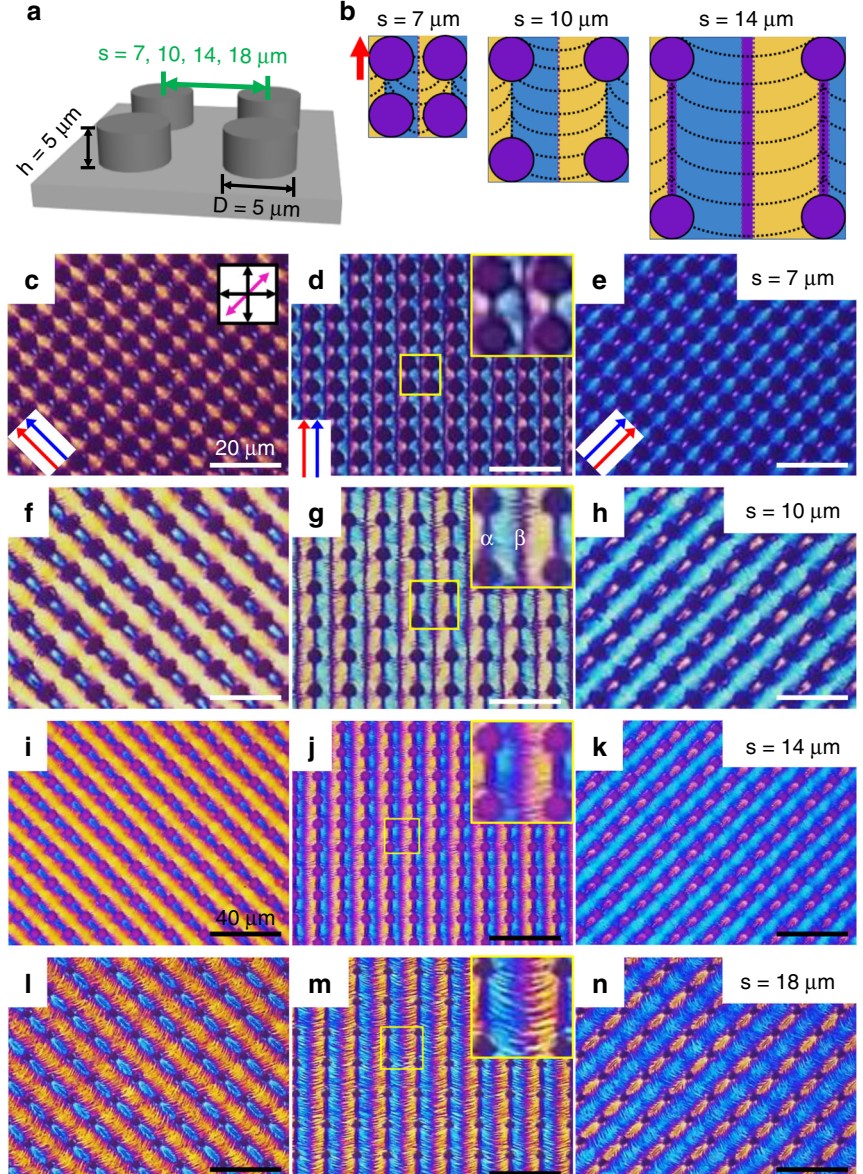

**Fig. 4** Variation in DNA microstructures depending on separation between microposts. **a** The micropost-patterned substrate and its dimensions. **b** Schematic sketch of the aligned DNA chains for each condition. **c–n** POM images of DNA microstructures for varying *s*, and by rotating samples ± 45°. White scale bars represent 20 μm and black ones are 40 μm

balanced collective behaviour of DNA elasticity and shearing effect, as described in the following equation (1) of elastic free energy, which was derived in the N phase[19,21],

$$f = \frac{K_1}{2}(\nabla \cdot \hat{n})^2 + \frac{K_2}{2}(\hat{n} \cdot (\nabla \times \hat{n}))^2 + \frac{K_3}{2}(\hat{n} \times (\nabla \times \hat{n}))^2 + \frac{E}{2}\frac{\partial \rho_p}{\rho_p^0},$$

(1)

where $K_1$, $K_2$ and $K_3$ are the elastic constants for splay, twist and bend deformation, respectively, and $\partial \rho$ is the variation in the local DNA density, $\rho^0$ is the mean DNA density, and $E$ is an elastic compressibility induced by dilative stress. In the given system, splay deformation is not usually found because $K_1 > K_2$, $K_3$[29,30] at high concentration or low temperature or high molecular weight, so bend deformation is dominant, resulting in $n_{DNA}$ aligned parallel to the contact line (Fig. 1b–e). As a result, the direction of lyotropic building blocks in the Col phase is parallel to the contact line to minimise free energy. But the receding behaviour near the contact line induces dilative and compressive stress in

the building blocks, generating buckling or undulation to compensate this energetically unstable state.

**Correlation with collective behaviour of lyotropic LC materials.** Microstructures made of cellulose nanocrystal (CNC) and sunset yellow (SSY) were investigated and were found to show almost identical and totally different morphologies, respectively (Supplementary Fig. 3). This can be explained by considering the aggregated shape of the molecules in the LC phase. In detail, DNA and CNC are made of chain-like long building blocks based on covalent bonds, whereas the SSY column is composed of very short discotic molecules stacked by noncovalent bonds (Supplementary Table 1).

In detail, lyotropic LC materials having the Col phase form a bend deformation parallel to the contact line during solvent evaporation. And then, two different phenomena can be observed as the contact line recedes, depending on the length of the building block. (1) The short building block-based lyotropic

LC materials, like SSY, form multiple domain walls owing to undulations, resulting from expansion and contraction near the interface, which look like cracks (Supplementary Fig. 3d–f)[26]. (2) This undulation is suppressed in the chain-based lyotropic LC polymers as the concentration increases[19], which is not observed in the highly concentrated condition[31]. This likely results in the differences among the three lyotropic LC materials during the formation of microstructures on the microposts, which cannot be easily explained by normal elastic energy under the steady state condition, as shown before[32].

To understand this, the collective behaviour of the lyotropic LC materials and their rheological properties should be discussed. First, we estimated the Deborah number ($De$) of the 25 mg/ml DNA solution by measuring the storage/loss modulus, which is smaller than unity, i.e., $De = (t_c/t_p) \sim 0.37$, where $t_c$ is the relaxation time measured by a rheometer and $t_p$ is scaled as $s/u$ ($s$ the separation distance and $u$ the pulling speed) (Supplementary Fig. 4a). Thus, in our platform the DNA solution is liquid-like. Viscosity and surface tension measurements (Supplementary Fig. 4b, c) were also carried out to determine the Capillary number $Ca(= \eta u/\sigma) \simeq O(10^{-3})$, where $\eta$ is the viscosity obtained from the rheometer measurement, $u$ is the pulling speed and $\sigma$ is the surface tension of the DNA solution determined by the pendant-drop method. Based on these data, we concluded that the capillary flow effect was dominant in our system. Accordingly, in this situation, the collective behaviour of the materials indicates that the chains can be stretched and/or aligned in some way due to capillary flow and/or contact line motion.

In addition, we performed dynamic viscosity measurements for the DNA, CNC and SSY, and obtained quite different results. Shear thinning behaviour (reduced viscosity with increasing shear rate) was observed in the DNA and CNC solutions (Supplementary Fig. 4b, d), whereas the SSY solution showed Newtonian behaviour (Supplementary Fig. 4d). This can indirectly explain why the SSY solution did not produce stable microstructures in our platform (Supplementary Fig. 3d–f). A Newtonian fluid

cannot produce collective behaviour due to the lack of elasticity effects. However, unfortunately, in this problem, it is extremely difficult to measure the exact elastic energy because their collective behaviour is motivated by the viscous drag effect, stretching the chains. These varieties of bio- or biocompatible material-based microstructures should also be of interest and further investigated because each material exhibits unique characteristics, as well as hydrodynamic interaction, in the given condition.

**Potential application of the DNA microstructures**. To show the usefulness of our platform we fabricated well-oriented plasmonic gold nanorods (GNRs) in DNA medium (Fig. 5). The extinction colours, blue and red of the DNA-GNR film were correlated with the longitudinal and transverse modes, resulting from the size of the GNR, which was 45 nm long and 15 nm in diameter (the inset in Fig. 5a)[22]. With varying angle, the extinction colours under a polarizer appeared bluish-grey (Fig. 5c), reddish-pink (Fig. 5e) and mixed colours (Fig. 5d, f). This suggests a way of using well addressed DNA microstructures as templating tools for GNRs or other guest functional materials, to enhance optical, magnetic and electronic properties.

## Discussion

We first report that microstructure arrays of DNA materials can be generated simply by varying the PD on topographic microposts. Competitive interaction in the meniscus-induced self-assembly on the microposts can change the orientation of the DNA chains. Comet-like and ice cream cone-like optical textures were directly observed using POM. This interesting morphogenesis provides a simple method for fabricating various kinds of microstructures using DNA material. The achievement also provides a method of using other anisotropic biomaterials, which are otherwise abundantly produced and pointlessly wasted, in

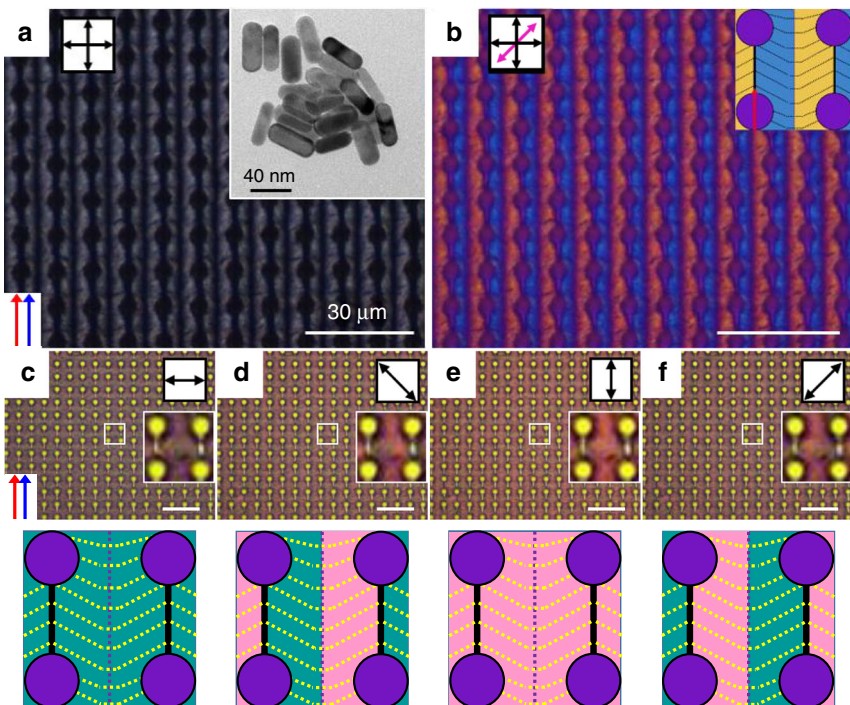

**Fig. 5** Plasmonic DNA–gold nanorods film fabricated on the microposts. **a**, **b** POM images without and with a first-order retardation plate. The inset TEM image shows the dimensions of the PEG-modified gold nanorods. **c**–**f** Optical colour textures depending on the angle between a polariser and pulling direction show blue **c**, red **e**, and mixed colour **d**, **f**. Black scale bar represent 40 nm and white ones are 30 μm

potential applications such as the plasmonic colour film shown in Fig. 5, when our result meets gold nanorods.

## Method

**Sample preparation and optical characterisation.** DNA (Deoxyribonucleic acid sodium salt from salmon testes, Sigma Aldrich) was dissolved in deionised water at 25 mg/ml without further purification. Pristine silicon, micropost-patterned silicon, or glass substrates were rinsed with acetone, ethanol and deionised water and then treated with $O_2$ plasma for 5 min to eliminate any organic/inorganic impurities.

For the doctor blade experiment, micropost-patterned silicon and glass substrates were sandwiched with a gap using an aluminium foil of 40 μm thickness. The DNA solution was injected into the sandwich cell by capillary forces, and then the upper glass piece of the sandwich cell was pulled in the heating stage (Linkam TST350). The temperature range and pulling speeds were 25 °C and 3 μm/s, respectively.

The optical textures of the fabricated DNA film were directly observed by POM (Nikon LV100POL) using a first-order retardation plate ($\lambda = 530$ nm). Fluorescent confocal polarising microscopy (C2 plus, Nikon) with a linearly polarised laser source ($\lambda = 488$ nm) was used to observe the meniscus line of the DNA microstructures. For this experiment, the DNA solution was mixed with a fluorescent dye molecule, Acridine orange (Aldrich).

## Data availability

The data that support the findings of this study are available from the corresponding author upon reasonable request.

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

## Acknowledgements

We thank Professor Joonwoo Jeong in UNIST for helpful discussions. This work was supported by a grant from the National Research Foundation (NRF) and was funded by the Korean Government (MSIT) (2017R1E1A1A01072798, 2018R1A5A1025208, 2017M3C1A3013923, and 2018R1C1B6004190).

## Author contribution

Y.J.C and S.M.P. contributed equally to this work. Y.J.C and D.K.Y. designed the research; Y.J.C, S.M.P. and R.Y. performed the micropost experimental works; H.K. performed the rheology experiment and analysis, Y.J.C, S.M.P. and D.K.Y. analysed results and wrote the manuscript.

## Additional information

**Competing Interests:** The authors declare no competing interests.

