## [Peer Review File · Nature Communications]

Reviewers' comments:

Reviewer #1 (Remarks to the Author):

The article by Cha et al. presents interesting experimental observations on the diverse alignment patterns arising in long DNA filaments upon a combination of drying, combing and interaction with geometrical constraints provided by micro-pillars.

It raises more questions than it answers, which may be seen as a merit. However, while most of the experiments are rigorous, systematic and well-presented, I think the article mainly suffers from the lack of quantitative or at least qualitative discussion of the origin of the observed behavior, so that the claim of further applications and extension to other biomaterials can be put on firmer basis.

To this aim, I would strongly suggest the authors try to address, either by further experiments, or simulations or at least scaling arguments, the following questions:

- What is the role of elastic constants?

Specifically, most lyotropic LCs, coupled or not to self-assembly like PBLG or chromonics, display a twist constant about an order of magnitude lower than splay and bend ones. In ref. [21], the same authors found elastic anisotropy to play a role in the formation of zigzag structures in rectangular micro-channels. Can similar arguments hold in this case?

Furthermore, Ref. [27] explicitly addresses the different possible textures of nematic and columnar chromonic phases in micropost arrays, understanding them in terms of elastic constants. Which arguments can be translated to DNA, and which others cannot?

Finally, anisotropy of elastic constants was observed to promote symmetry breaking in chromonic tactoids in Tortora & Lavrentovich, PNAS 108, 5163 (2011), and later in refs. [28] and [29] in cylindrical channels.

The observed curvature of meniscus is related, in line 113, to such symmetry breaking, but the connection should be made more explicit in the text.

- What is the role of wettability and contact angle?

Have you measured the contact angle? Is it constant over time?

Have you tried to image the moving contact line at higher magnification? Is its motion smooth or discontinuous?

(How) will a change in boundary conditions (at substrate and/or pillars) affect the regimes?

- What is the role of DNA chirality – if any - in the observed behavior?

In shorter DNA strands, the chirality clearly shows up in the cholesteric phase: do you expect any implications for the observed patterns?

Further questions:

- Would it be possible to directly measure the orientation of DNA helices during pattern formation and at steady state?

Fluorescence anisotropy with acridine orange or ethidium bromide, for example, may provide direct assessment of the claimed director configuration in e.g. region γ in Fig. 2d and 3g.

- Are the samples completely dehydrated, or kept at controlled humidity? Are the patterns stable over time, or do cracks appear upon further dehydration?

- Does the pulling speed affect the behavior? Was the 3 $\mu\text{m/s}$ value chosen to fit the drying kinetics or for what other reasons?

Some additional remarks:

- In line 56, methods to orient DNA should also include magnetic ordering (e.g. associated to drying in Morii et al. Biomacromolecules 5, 2297 (2004))

- In lines 109, 111 and 144 there are missing letters referring to Figures.

- While in the text (line 95) an angle of 22.5° is cited, Figure 2 reports 67.5° . Please unify the convention.

- Reference 30 is not cited throughout the text.

- The meaning of lines 160-161 is obscure, likely due to missing or too many words.

Reviewer #2 (Remarks to the Author):

This investigation details the study of drops comprised of a suspension of DNA, drying over a substrate with cylindrical posts. The posts serve as pinning sites for the contact line of the droplet to be shaped, ultimately affecting the way the DNA is ordered on the substrate as the droplet evaporates. Drying droplets as a method of templating has been broadly used with materials from nanoparticles/nanocrystals to polymers, but this example is interesting because it takes advantage of features on a substrate to guide the drying process, which ultimately affects the resulting structures obtained. However, the experiments could be more thorough, and the paper could be better motivated and framed within the existing body of work that uses drying droplets as a means of ordering materials. I could recommend the paper for publication after a few more experiments and if the following comments are addressed (some are minor):

In the abstract, the phrase, "finite feature size," is vague. What does this refer to exactly?

"What should be noted here is that the meniscus lines are not tight but curved." – Is "tight" the intended word here? Do you mean "straight" instead?

How are the schematics in Figure 3 made? Is it merely a guess as what the shape of the meniscus is? The contact angle (both advancing and receding contact angles with this specific substrate material) is surely important in shaping the meniscus. How is it determined that the meniscus is curved in one case but straight for the other? The cited works do not automatically support this claim. There is also an error in the paper for which citations are intended for this part of the paper. It seems that the intended citations for this is 30 and 31, not 28 and 29. Are the topographical posts used in this study like those used in the studies of either 30 or 31, in composition or geometry? I recommend doing more careful measurements of the receding and advancing contact angles and of the meniscus shape, since these details are important for the resulting morphology of the DNA. Using optical microscopy to observe just the evolution of the contact line in the xy-plane, without more careful measurements of the contact angle (information in the z-direction), is not enough to support the conclusions drawn in the text. The microscope images are also of a poor quality (blurry, out-of-focus), making it difficult to discern the evolution of the contact line in the xy-plane from just the data provided in the figure.

At the end of the discussion, you also mention that the DNA undulates, and you cite a few previous works that also show this phenomenon. But what is the mechanical origin of these undulations? The drying behavior and resultant structures of other lyotropic liquid crystal droplets has also been previously investigated by Davidson et. al. (Nature Communications, 2017, (8)15642), and a lot of the descriptions and mechanisms detailed in that work can help in understanding some of aspects of the patterns obtained in this study, such as the undulating texture. An exact mechanism is not needed, but at least mentioned from reviewing other works would be useful for readers.

Finally, in the conclusion, it is stated that the microstructures of DNA generated have potential applications, but what these applications are is not clear to me. How exactly does this technique prevent the DNA from being "wasted"? When DNA is purified, it is isolated from other components, such as cell debris, within a solution. This method of templating the DNA is interesting, but does not address at all how DNA is isolated in solution, nor is it clear to me how the purity of the DNA is at all improved, post-processing. What IS clear to me is how being able to align polymeric materials, more generally, is useful for many applications, such as for generating plastics with anisotropic properties. This study is an interesting method of this that could be applied to many other systems, but how the paper is currently motivated, in the abstract, the introduction, and the conclusion, is unconvincing to me. I suggest expounding upon the motivation of this work to better frame the investigation within the context of other, existing works.

For instance, drying as a method of templating has been used broadly, with many types of materials as well. One prominent example is the templating of nanocrystals by Dong et. al. (Nano Lett., 2011, 11 (2) 841-846). A brief mention in the introduction of other works that have used drying droplets as a method of templating would be appropriate and would help to put this investigation into a broader, more applicable context.

Reviewer #1:

The article by Cha et al. presents interesting experimental observations on the diverse alignment patterns arising in long DNA filaments upon a combination of drying, combing and interaction with geometrical constraints provided by micro-pillars. It raises more questions than it answers, which may be seen as a merit. However, while most of the experiments are rigorous, systematic and well-presented, I think the article mainly suffers from the lack of quantitative or at least qualitative discussion of the origin of the observed behavior, so that the claim of further applications and extension to other biomaterials can be put on firmer basis. To this aim, I would strongly suggest the authors try to address, either by further experiments, or simulations or at least scaling arguments, the following questions:

Thank you for the kind comments and we tried to revise the manuscript as recommended. Especially, we conducted the experiments with the other biomaterials, e.g. microstructures of cellulose nanocrystal (CNC), which are now added in supporting information (Supplementary Figure 3), showing almost the identical result with DNA's. On the contrary to this, sunset yellow (SSY) based on experiments does not show the similar result, which might be related with the persist length difference to form microstructures. These are added in pages 8 and 9 of the revised manuscript.

Q1. What is the role of elastic constants?

Specifically, most lyotropic LCs, coupled or not to self-assembly like PBLG or chromonics, display a twist constant about an order of magnitude lower than splay and bend ones. In ref. [21], the same authors found elastic anisotropy to play a role in the formation of zigzag structures in rectangular microchannels. Can similar arguments hold in this case? Furthermore, Ref. [27] explicitly addresses the different possible textures of nematic and columnar chromonic phases in micropost arrays, understanding them in terms of elastic constants. Which arguments can be translated to DNA, and which others cannot? Finally, anisotropy of elastic constants was observed to promote symmetry breaking in chromonic tactoids in Tortora & Lavrentovich, PNAS 108, 5163 (2011), and later in refs. [28] and [29] in cylindrical channels. The observed curvature of meniscus is related, in line 113, to such symmetry breaking, but the connection should be made more explicit in the text.

A1. Thank you for the comment. As recommended, we changed these.

The experiment in Ref. [27] was conducted in a kind of “steady state”, in which hexadecane was used to “seal” the lyotropic chromonic LC (LCLC) not to evaporate water to maintain the LC phase. In this system, the elastic constant determines the structure so that K_3/K_1 is very important, and the hydrodynamic effect is not a dominant factor, which is totally different with our system. Indeed, as added in the supporting information, SSY based experimental results does not show the similar result with the previous one (Supplementary Figure 2), and thus the explanation in Refs. (27, 28, 29, and Tortora & Lavrentovich, PNAS 108, 5163 (2011)) cannot be identically applied to our system although it possibly has some minor relationship, which is found in Figure 4. Especially, in large separation (s) cases (Figure 4f-n), zigzag structures are generated, in which elastic properties should be considered and we tried to explain the details in the revised manuscript (pages 8-9). This is once again explained in the reviewer #2’s 3rd comment.

For the curvature of meniscus, we conducted additional experiments to show the curvature more clearly (Figure 3) and illustrate new schematic sketches based on the new experiments.

Q2. What is the role of wettability and contact angle?

Have you measured the contact angle? Is it constant over time?

Have you tried to image the moving contact line at higher magnification? Is its motion smooth or discontinuous? (How) will a change in boundary conditions (at substrate and/or pillars) affect the regimes?

A2. Thank you for the comment. As the reviewer suggested, we conducted the additional experiments with DNA solution like the below, now showing dramatic changes during the water evaporation although we could see the discontinuous motion in the revised Figure 3, which was added in pages 6 and 7.

Contact angle measurement results of (a) and (b) are for (c-e) and (f-h), respectively. (c-e) Contact angle changes of DNA solution-droplet (5 μl in volume at 25 mg/ml concentration) on a bare Si substrate during the water evaporation. (f-h) Contact angle changes of DNA solution-droplet (5 μl in volume at 25 mg/ml concentration) on microposts ($s = 10 \mu\text{m}$, $D = 5 \mu\text{m}$, and $h = 5 \mu\text{m}$) during the water evaporation.

Q3. What is the role of DNA chirality – if any - in the observed behavior?

In shorter DNA strands, the chirality clearly shows up in the cholesteric phase: do you expect any implications for the observed patterns?

A3. We performed the experiment many times but could not see the specific behavior for the chirality but we observe the symmetrical micro-structures. The almost identical behavior was found in the CNC experiments but not in the SSY case. In the shorter DNA case, we have not conducted the experiments at the moment because the material is too expensive but we can imagine the final structure would be like the current one based on the previous study performed in the microchannels (Yoon, D. K. et al, "Alignment of the Columnar Liquid Crystal Phase of Nano-DNA by Confinement in Channels", *Liq. Cryst.* 39, 571–577 (2012)). However, the topological defect induced by the specific chiral moiety was found in the other lyotropic LC material depending on the chiral dopant in the totally different experimental platform so we did not add this result in the reply or the revised manuscript, which will be reported in the near future work soon.

Q4. Would it be possible to directly measure the orientation of DNA helices during pattern formation and at steady state?

A4. During the pattern formation it is very hard to see the dynamic change from the measurement image, especially using water-immersed AFM, but we could see the straight- or wavy-line textures after

forming DNA microstructures like the below.

Q5. Fluorescence anisotropy with acridine orange or ethidium bromide, for example, may provide direct assessment of the claimed director configuration in e.g. region γ in Fig. 2d and 3g.

A5. As recommended, we conducted these experiments for tens of times because the experimental condition was very delicate, but we could not observe the high resolution images directly in the side views (α , β regions in (b,c)) and top view (γ region in (d,e)), which are provided in the figure below.

Instead, we estimated the Deborah number of 25 mg/ml DNA solution by measuring storage/loss modulus, which is smaller than unity, i.e. $De = (t_c / t_p) \sim 0.37$, where t_c is the relaxation time obtained from the rheometer measurement and t_p is scaled as s/u (s the separation distance and u the pulling speed) (f). Thus, the DNA solution is liquid-like at the relatively slow pulling speed.

The surface tension measurements of shear rate-dependent viscosity (g) and surface tension (h) using 25 mg/ml-DNA solution are carried out to show the Capillary number, $Ca (= \eta u / \sigma) \approx O(10^{-3})$, where η is viscosity obtained from rheometer measurement, u is the pulling speed and σ is surface tension of the DNA solution from the pendant-drop method. Thus, we concluded that the capillary force is dominant in our platform.

Additionally, we found the shear thinning behavior (which shows the reduced viscosity as increasing shear rate) (g), which means DNA chains may be stretched due to the viscous drag force. Therefore, in our experimental condition, the viscosity mainly plays a role to stretch DNA chains even in isotropic γ region in (d,e) through the evaporation-induced capillary force.

We did not add this result in the revised manuscript at this moment. The reason is that we also need to perform a systematic study by varying various parameters including viscosity, concentration of DNA, different length of DNA, and so on. We will execute more systematical studies in the near future.

Q6. Are the samples completely dehydrated, or kept at controlled humidity? Are the patterns stable over time, or do cracks appear upon further dehydration?

A6. The sample was completely dried for a few days, which showed the identical optical images as shown in the manuscript. And the patterns are very stable in the wide range of temperatures, 10 °C to 60 °C once they are dehydrated.

Q7. Does the pulling speed affect the behavior? Was the 3 $\mu\text{m/s}$ value chosen to fit the drying kinetics or for what other reasons?

A7. Thanks for the kind comment. We choose the 3 $\mu\text{m/s}$ -pulling speed to make the reliable and micropost-dependent morphologies. If the pulling speed is much slower than 3 $\mu\text{m/s}$, the intrinsic evaporation of water is dominant not to see the shear-dependent morphogenesis. On the contrary, the faster pulling speed than 10 $\mu\text{m/s}$ induces the splay deformation of DNA chains or only shear oriented DNA chains, which is also

consistent with the rheological property of the DNA solution. This is added in supplementary Figure 1, which is also revised in page 5.

Q8. Some additional remarks:

- In line 56, methods to orient DNA should also include magnetic ordering (e.g. associated to drying in Morii et al. *Biomacromolecules* 5, 2297 (2004))
- In lines 109, 111 and 144 there are missing letters referring to Figures.
- While in the text (line 95) an angle of 22.5° is cited, Figure 2 reports 67.5° . Please unify the convention.
- Reference 30 is not cited throughout the text.
- The meaning of lines 160-161 is obscure, likely due to missing or too many words.

A8. Thank you for the kind comment, we correct these.

Reviewer #2

This investigation details the study of drops comprised of a suspension of DNA, drying over a substrate with cylindrical posts. The posts serve as pinning sites for the contact line of the droplet to be shaped, ultimately affecting the way the DNA is ordered on the substrate as the droplet evaporates. Drying droplets as a method of templating has been broadly used with materials from nanoparticles/nanocrystals to polymers, but this example is interesting because it takes advantage of features on a substrate to guide the drying process, which ultimately affects the resulting structures obtained. However, the experiments could be more thorough, and the paper could be better motivated and framed within the existing body of work that uses drying droplets as a means of ordering materials. I could recommend the paper for publication after a few more experiments and if the following comments are addressed (some are minor):

- We appreciate the valuable comments and tried to revise manuscript as the reviewer suggested. Especially, we added the elastic energy terms and meniscus lines to clarify our result in the revised manuscript.

Q1. In the abstract, the phrase, “finite feature size,” is vague. What does this refer to exactly? “What should be noted here is that the meniscus lines are not tight but curved.” – Is “tight” the intended word here? Do you mean “straight” instead?

A1. Thank you for the comment. There were some typos and “finite” is changed to “fine”. We revised the manuscript properly according to the reviewer’s comments.

Q2. How are the schematics in Figure 3 made? Is it merely a guess as what the shape of the meniscus is? The contact angle (both advancing and receding contact angles with this specific substrate material) is surely important in shaping the meniscus. How is it determined that the meniscus is curved in one case but straight for the other? The cited works do not automatically support this claim. There is also an error in the paper for which citations are intended for this part of the paper. It seems that the intended citations for this is 30 and 31, not 28 and 29. Are the topographical posts used in this study like those used in the studies of either 30 or 31, in composition or geometry? I recommend doing more careful measurements of the receding and advancing contact angles and of the meniscus shape, since these details are important for the resulting morphology of the DNA. Using optical microscopy to observe just the evolution of the contact line in the xy-plane, without more careful measurements of the contact

angle (information in the z-direction), is not enough to support the conclusions drawn in the text. The microscope images are also of a poor quality (blurry, out-of-focus), making it difficult to discern the evolution of the contact line in the xy-plane from just the data provided in the figure.

A2. Thank you for the detailed comments. First of all, we conducted the contact angle measurements and tried to see the contact line directly by adding a fluorescent dye. The apparent receding angle-related meniscus is extremely difficult due to the very small structure (see the figure below) because our system is based on the simply pulling upper glass. However, we agree that this could be a good idea to investigate the dewetting patterns and we will try to set up new measurement device. Anyway, to improve our manuscript, the schematics were reconsidered and updated, please see the revised Figure 3 and a Figure in A5 for the reviewer #1's comment (please see page 5 in this reply). We also changed and added references as the referee suggested.

The high-resolution images measured in the additional experiments show the convex contact line at $\phi = 0^\circ$, 45° as shown in the revised Figure 3, revealing the different meniscus-moving speed on the microposts.

Q3. At the end of the discussion, you also mention that the DNA undulates, and you cite a few previous works that also show this phenomenon. But what is the mechanical origin of these undulations? (1) The drying behavior and resultant structures of other lyotropic liquid crystal droplets has also been previously investigated by Davidson et. al. (Nature Communications, 2017, (8)15642), and a lot of the descriptions and mechanisms detailed in that work can help in understanding some of aspects of the patterns obtained in this study, such as the undulating texture. An exact mechanism is not needed, but at least mentioned from reviewing other works would be useful for readers.

A3. Thank you for the comment. As recommended, we added new references and revised the manuscript

(pages 8-9).

References:

- a) Z. S. Davidson et al. (Nature Communication, 2017 (8) 15642), (Ref. [26] in revised paper)
- b) I. I. Smalyukh et al. (Physical Review Letter, 2006 (98) 177801), (Ref. [19] in revised paper)
- c) Y. J. Cha et al. (Advanced Materials 2017, (29) 1604247) (Ref. [21] in revised paper)

In these works, (a) sunset yellow (SSY), (b) Lambda-DNA, and (c) salmon sperm DNA were used in (a,b) simple droplet evaporation and (c) shear-induced film condition. As it is well-known, these building blocks in a highly concentrated solution condition show lyotropic LC phase, especially columnar (Col) phase. In this condition, the bend elastic constant (K_3) is smaller than the splay constant (K_1), so the director of lyotropic building blocks in Col phase is parallel with the contact line to minimize the free energy. But the receding behavior near the contact line induces the dilative and compressive stress of the building blocks to generate buckling or undulation to compensate this energetically unstable state.

Q4. Finally, in the conclusion, it is stated that the microstructures of DNA generated have potential applications, but what these applications are is not clear to me. How exactly does this technique prevent the DNA from being “wasted”? When DNA is purified, it is isolated from other components, such as cell debris, within a solution. This method of templating the DNA is interesting, but does not address at all how DNA is isolated in solution, nor is it clear to me how the purity of the DNA is at all improved, post-processing. What IS clear to me is how being able to align polymeric materials, more generally, is useful for many applications, such as for generating plastics with anisotropic properties. This study is an interesting method of this that could be applied to many other systems, but how the paper is currently motivated, in the abstract, the introduction, and the conclusion, is unconvincing to me. I suggest expounding upon the motivation of this work to better frame the investigation within the context of other, existing works.

For instance, drying as a method of templating has been used broadly, with many types of materials as well. One prominent example is the templating of nanocrystals by Dong et. al. (Nano Lett., 2011, 11 (2) 841-846). A brief mention in the introduction of other works that have used drying droplets as a method of templating would be appropriate and would help the put this investigation into a broader, more applicable context.

A4. Thanks for the kind comments. We are also pretty much interested in the application of patterned DNA microstructures as suggested in DNA-based LC display, DNA templated LC alignment, and DNA-mediated nanoparticle arrays for the display applications. Here, we added Supplementary Figure 4 to show how our interesting structure is used for the plasmonic color film like the below.

Reviewers' comments:

Reviewer #1 (Remarks to the Author):

Thank you for the detailed answers to my comments and questions. Overall, the answers are quite convincing, but my main concern is that the new manuscript only conveys a very limited amount of the new experiments and discussions, and would therefore raise the same questions to the readers, without properly addressing them.

Specifically:

- Extension to other biomaterials: no explanation is even tried in the text for the fact that nanocellulose displays similar behaviour to DNA, while the chromonic LC doesn't.
- A5 and answer to referee 2 about the shape of the meniscus: the new figure 3 (b-d and f-h) is not really clearer than before in showing the z-shape of the meniscus; why not to show the fluorescence image reproduced in the rebuttal letter, even if not outstanding?
- A2&A5 on surface tension: what should we conclude from the surface tension measurements? If this and viscosity are used to estimate the Capillary number and thus support an explanation for the observed behaviour, such measurements should be included at least in the supplementary and such explanation should be included.
- Moreover, on A7 and pulling speed: how does the "magic" 3 $\mu\text{m}/\text{sec}$ value depend on the measured quantities? Again, in the text there is no trace of an explanation.

Reviewer #2 (Remarks to the Author):

The authors have addressed the majority of my comments, and I appreciate the additional experiments that the authors have performed in response. The additional experiments greatly improve the manuscript. However, some of the edits to the manuscript need to be addressed, and warrant further examination.

The work cited for claiming that $K_1 \gg K_2, K_3$ in the authors' system, citation 29, measures the elastic constants for sunset yellow, and not DNA. Even more, the morphologies seen for sunset yellow are starkly different from those seen for DNA. Furthermore, in [29], K_1 was indeed measured for sunset yellow, and was found to be 10 pN, with $K_2 = 0.6$ pN and $K_3 = 10$ pN. This citation clearly does not apply to this claim. Citation 21 does state this, however, but for their specific system, in which their DNA contour length is around 16.3 microns. This is orders of magnitude larger than the contour length of the DNA used in your experiments, which is 680 nm. How are you sure that this statement applies to your system? Please argue this claim more carefully and check your citations carefully. This argument in the text needs to be explored further and more thoroughly, especially given the large differences seen in sunset yellow compared to the DNA and even the cellulose nanocrystals system.

Perhaps more detailed experiments for this are outside of the scope of the current manuscript, but at least, the open questions should be brought out more in the text, and if possible, a closer examination of the elastic properties of these 3 materials, explicitly listing their similarities and differences. I recommend that the authors carefully compare system length scales and elastic constants to make a more thorough and cohesive argument. That cellulose nanocrystals form similar patterns is interesting - what are the length scales there? Are there elastic constant measurements in the literature for these materials?

It is also jarring to have new information given in the conclusion. I am addressing specifically the experiments of DNA with gold nanorods. This experiment does hint at more possible applications of these experiments, and including them is indeed a large improvement from the first draft of the manuscript and better motivates the work. However, it does not do the experiments justice to have them thrown into the conclusion. I think the manuscript would be much improved if this was delved into more and included in the main discussion section of the manuscript.

After these points are addressed to satisfaction, then this manuscript would merit publication.

Reviewer #1:

Thank you for the detailed answers to my comments and questions. Overall, the answers are quite convincing, but my main concern is that the new manuscript only conveys a very limited amount of the new experiments and discussions, and would therefore raise the same questions to the readers, without properly addressing them.

> Thank you very much for the kind comment. We tried to revise the manuscript to clarify the contents as suggested.

- Extension to other biomaterials: no explanation is even tried in the text for the fact that nanocellulose displays similar behaviour to DNA, while the chromonic LC doesn't.

> This may be explained by the molecular assembly and viscous behavior of the materials and added in page 9 like the below.

The similar microstructures made of cellulose nanocrystal (CNC) and sunset yellow (SSY) are investigated to show almost identical and totally different morphologies, respectively (Supplementary Fig. 3). This can be simply understood considering the aggregated shape in each case. In detail, DNA and CNC are made of chain-like building blocks, while SSY is composed of discotic molecule. This possibly changes the physical phenomena in terms of "stickiness" during the formation of microstructure on the microposts, which cannot be easily explained with the normal elastic energy under the steady state as shown before³¹. To understand the exact mechanism of the structure formation, all the individual materials have to be fully investigated from the extensive experiments, which are out of the current research scope. Nonetheless, these varieties of bio- or biocompatible material-based microstructures should also be of interest because each material has the unique elastic characteristic as well as hydrodynamic interaction in the given condition.

- A5 and answer to referee 2 about the shape of the meniscus: the new figure 3 (b-d and f-h) is not really clearer than before in showing the z-shape of the meniscus; why not to show the fluorescence image reproduced in the rebuttal letter, even if not outstanding?

> We appreciate your suggestion and added the fluorescent confocal images in Figure 3k and l. We hope that the new figures are helpful for researchers to understand the shape of the meniscus.

- A2&A5 on surface tension: what should we conclude from the surface tension measurements?

If this and viscosity are used to estimate the Capillary number and thus support an explanation for the observed behaviour, such measurements should be included at least in the supplementary and such explanation should be included.

> We appreciate your comment, and missed some points. Now the detailed explanations are added in page 9 like the below.

In terms of this view point, we need to understand the mechanical properties of collective DNA chains. For this, we estimated the Deborah number (De) of 25 mg/ml DNA solution by measuring storage/loss modulus, which is smaller than unity, i.e. $De = (t_c / t_p) \sim 0.37$, where t_c is the relaxation time measured

by a rheometer and t_p is scaled as s/u (s the separation distance and u the pulling speed) (Supplementary Fig. 4a). Thus, the DNA solution is liquid-like in our platform, meaning viscosity is important rather than elasticity. The surface tension measurements (Supplementary Fig. 4b, c) using 25 mg/ml-DNA solution were also carried out to show the Capillary number, $Ca (= \eta u / \sigma) \approx O(10^{-3})$, where η is viscosity obtained from rheometer measurement, u is the pulling speed and σ is surface tension of the DNA solution from the pendant-drop method. Thus, we concluded that the capillary force is dominant in our platform. Additionally, we also found the shear thinning behaviour (which shows the reduced viscosity as increasing shear rate) (Supplementary Fig. 4b), which means DNA chains may be stretched due to the viscous drag force, which makes hard to measure the exact elastic energy.

- Moreover, on A7 and pulling speed:

how does the "magic" 3 $\mu\text{m}/\text{sec}$ value depend on the measured quantities? Again, in the text there is no trace of an explanation.

> In fact, it is not the magic number. The speed value can be roughly estimated from the Deborah number. From the rheology measurement, we obtained the relaxation time $t_c \approx 0.82$. To obtain the DNA structure, the solution should behave like liquid. So, the flow speed has to satisfy the below relation,

$$u < s/t_c \sim 10^{-5}/0.82 \text{ [m/s]} \approx 12 \text{ [\mu m/s]}.$$

Additionally, we varied the pulling speeds near the critical speed based on the rough model. From multiple experiments, we have found out that 3 to 7 $\mu\text{m}/\text{s}$ made the reliable microstructures, while below 3 $\mu\text{m}/\text{sec}$ -pulling speed the irregular structures were formed. The faster made interesting structures but the reliable or microposts-dependent morphologies were not generated (Supplementary Fig. 1). Therefore, the pulling speed is fixed at 3 $\mu\text{m}/\text{s}$ in this study. We still believe that our experimental observations could sparkle some interests in this area and offer a new physicochemical avenue for potential patterning applications using DNA although we performed a limited amount of the experiments. Overall, the manuscript was revised in page 5 like the below.

In the beginning of this work, various pulling speeds were tested and 3 to 7 $\mu\text{m}/\text{s}$ made the reliable microstructures, while the slower did not form the regular structures. The faster made interesting structures but the reliable or microposts-dependent morphologies were not generated (Supplementary Fig. 1). Therefore, the pulling speed is fixed at 3 $\mu\text{m}/\text{s}$ in this study.

Reviewer #2 (Remarks to the Author):

The authors have addressed the majority of my comments, and I appreciate the additional experiments that the authors have performed in response. The additional experiments greatly improve the manuscript. However, some of the edits to the manuscript need to be addressed, and warrant further examination.

> First of all, we are pleased to see the Referee #2's note that "The authors have addressed the majority of my comments, and I appreciate the additional experiments that the authors have performed in response. The additional experiments greatly improve the manuscript." We also thank the Referee #2 for the further comments. In particular, we'd like to thank you for the comment for the elastic energy part, and tried to explain it, and now it's much clearer than before. We did our best to clarify following comments, one by one.

The work cited for claiming that $K_1 \gg K_2, K_3$ in the authors' system, citation 29, measures the elastic constants for sunset yellow, and not DNA. Even more, the morphologies seen for sunset yellow are starkly different from those seen for DNA. Furthermore, in [29], K_1 was indeed measured for sunset yellow, and was found to be 10 pN, with $K_2 = 0.6$ pN and $K_3 = 10$ pN. This citation clearly does not apply to this claim.

> In reference [29], Prof. Lavrentovich and his colleagues explained the elastic constant and the correlation between contour and persistence length semiflexible lyotropic polymeric LC. So, we think the reference can be used to explain the elastic property of DNA chain in our system. FYI, in [29], K_1 was indeed measured for sunset yellow, and was found to be 10 pN, with $K_2 = 0.6$ pN and $K_3 = 10$ pN. This citation clearly does not apply to this claim. Actually, in reference [29], the elastic constants of SSY are $K_1 = 4.3 \pm 0.4$, $K_2 = 0.7 \pm 0.07$, and $K_3 = 6.1 \pm 0.6$, and those of poly(γ -benzyl glutamate) are $K_1 = 10$ pN, $K_2 = 0.6$ pN and $K_3 = 10$ pN.

Citation 21 does state this, however, but for their specific system, in which their DNA contour length is around 16.3 microns. This is orders of magnitude larger than the contour length of the DNA used in your experiments, which is 680 nm. How are you sure that this statement applies to your system? Please argue this claim more carefully and check your citations carefully. This argument in the text needs to be explored further and more thoroughly, especially given the large differences seen in sunset yellow compared to the DNA and even the cellulose nanocrystals system.

> We appreciate your comments. We deeply thought about it and had struggled to figure it out. And we concluded that the shape of building blocks and viscous behavior is important in our system as explained in reviewer 1's comment and our response. By the way, in the steady state, the contour length is important as you mentioned because the shorter can make the LC phases than longer one as explained by Boulder group [ref. 15], 6-20 bp-DNA showed the LC phases under the Onsager line. So, the shorter DNA (though ours are not comparable with the shortness in Boulder group's work) is useful to form the LC phase to study phase behaviour. However, once again, in our platform, the elastic term is less important than the viscous property of DNA solution. Now the detailed explanations are revised in page 9.

Material	Persistence length (λ_p)	Contour length (L)	Diameter (D)	Aspect ratio (L/D) + (λ_p/D)	Flexibility (λ_p/L)
Salmon sperm DNA (DNA)	50nm	680nm	2nm	340(25)	Semiflexible ($\ll 1$)
Sunset Yellow FCF (SSY)	10nm (*Nematic)	? nm (≈ 10 nm)	1nm	$\approx 10(10)$	Semiflexible (< 1)
Cellulose nanocrystal (CNC)	150nm	300nm	10nm	30(15)	Semiflexible (< 1)

The above is the table to show the material properties known and references are like the below.

SSY: ref. [29] Zhou S. et al. Elasticity of lyotropic chromonic liquid crystals probed by director reorientation in a magnetic field. *Phys. Rev. Lett.* 109, 037801 (2012).

CNC: ref. [31] Iwamoto, S., Lee, S. & Endo, T. Relationship between aspect ratio and suspension viscosity of wood cellulose nanofibers. *Polymer J.* 46, 73-76 (2014)

Based on this comparison, our behavior cannot be explained because the elastic property between CNC and DNA is different though the behavior in our system is similar.

Perhaps more detailed experiments for this are outside of the scope of the current manuscript, but at least, the open questions should be brought out more in the text, and if possible, a closer examination of the elastic properties of these 3 materials, explicitly listing their similarities and differences. I recommend that the authors carefully compare system length scales and elastic constants to make a more thorough and cohesive argument. That cellulose nanocrystals form similar patterns is interesting - what are the length scales there? Are there elastic constant measurements in the literature for these materials?

> We extensively reviewed many literatures and concluded it like the just above. At this point, we believe that the current observation could sparkle some interests in this area and offer a new physicochemical avenue for potential patterning applications using DNA. To clarify this point, we revised the resubmitted version of the manuscript (please see page 9).

It is also jarring to have new information given in the conclusion. I am addressing specifically the experiments of DNA with gold nanorods. This experiment does hint at more possible applications of these experiments, and including them is indeed a large improvement from the first draft of the manuscript and better motivates the work. However, it does not do the experiments justice to have them thrown into the conclusion. I think the manuscript would be much improved if this was delved into more and included in the main discussion section of the manuscript.

> We appreciate your comment and now Fig. 5 was added to show the DNA-gold nanorods experimental result.

Reviewers' comments:

Reviewer #1 (Remarks to the Author):

I appreciate the further effort made by the authors to provide more experiments and a clearer discussion.

Still, I don't see what "stickiness" (page 9) can mean in the context of the different materials they explore; do the authors mean interaction with the substrate and tendency to align?

Quantities affecting instabilities and pattern formation certainly include elastic constants, viscosity (or viscosities) and surface tension; on their turn, such quantities also depend on molecular or sovramolecular scale properties.

One would expect the authors to measure viscoelastic properties - or get them from literature - for the different materials, in order to discuss their different behaviour. Instead, they simply compare molecular properties.

In any case, in the rebuttal letter's table, the aspect ratio defined as contour length over diameter for long semiflexible polymers simply makes no sense, because the persistence length is dominant. Furthermore, at the investigated concentration, DNA filaments are entangled with mesh size few hundreds of nm (simply estimated from the elastic modulus plateau).

Therefore, authors should be more careful when identifying the key length scales and parameters.

Overall, my conclusion is that the article can be published after further cleaning of English and more rigorous use of physical terms.

Reviewer #2 (Remarks to the Author):

The manuscript is much improved, especially from the initial submission. However, more description of the physical mechanism behind how $K_1 \gg K_2, K_3$ in your system would add to the quality of the paper, as mentioned in my last review (just a sentence or two). I also meant citation [19] by Smalyukh et. al. and not citation [21] in my previous review, but this is a minor mistake. I appreciate the authors looking more into the elastic energy argument, comparing the DNA system to CNC and SSY.

The language for the description of why CNC and your DNA system have similar behaviors, but not SSY, should be softened. In your revised manuscript, you write:

"This can be simply understood considering the aggregated shape in each case. In detail, DNA and CNC are made of chain-like building blocks, while SSY is composed of discotic molecule. This possibly changes the physical phenomena in terms of "stickiness" during the formation of microstructure on the microposts, which cannot be easily explained with the normal elastic energy under the steady state as shown before."

It is better to make clear that this is a hypothesis, since you do not have direct experimental support of this. Since the elastic constant argument does not seem to explain the observed phenomena, the aggregated shape could be one possible explanation, but further experiments, outside the scope of the paper, are needed. Your chart detailing the common characteristics between the 3 materials that you included in the rebuttal should be included in supplementary materials and should be referenced in your description of why the elastic energy argument fails to describe what you observe. An additional few sentences detailing the elastic energy argument and why it fails is also needed.

The inclusion of Figure 5 in the text is a huge improvement to the manuscript. However, the color switching in Figure 5, c-f is not so obvious in the micrographs. Perhaps zooming in to the structure, instead of having a zoomed out view, will help make this more evident.

I appreciate all of the work the authors have done to improve the manuscript thus far. After these additional, minor revisions, I can recommend publication of this article.

Reviewers' comments:

Reviewer #1 (Remarks to the Author):

I appreciate the further effort made by the authors to provide more experiments and a clearer discussion. Still, I don't see what "stickiness" (page 9) can mean in the context of the different materials they explore; do the authors mean interaction with the substrate and tendency to align?

Quantities affecting instabilities and pattern formation certainly include elastic constants, viscosity (or viscosities) and surface tension; on their turn, such quantities also depend on molecular or supramolecular scale properties.

One would expect the authors to measure viscoelastic properties - or get them from literature - for the different materials, in order to discuss their different behaviour. Instead, they simply compare molecular properties.

In any case, in the rebuttal letter's table, the aspect ratio defined as contour length over diameter for long semiflexible polymers simply makes no sense, because the persistence length is dominant. Furthermore, at the investigated concentration, DNA filaments are entangled with mesh size few hundreds of nm (simply estimated from the elastic modulus plateau).

Therefore, authors should be more careful when identifying the key length scales and parameters.

> I agree with your opinion in our case, though the nanoDNA can show the different phase behavior when the contour length is very short as shown in 6-20 mer nano(or oligo)DNA [Ref 15]. Anyway, the bend elastic term, K_3 enlarges as persistence length increases. SSY made of supramolecular discotic molecules can form domain walls (look like cracks in POM images) after the aggregation (stacked discotic molecules) is broken when the bend deformation is large as shown in [Ref 26]. However, this cannot be happened in polymeric lyotropic LCs such as DNA and CNC. Considering this point, we revised the manuscript like the below in page 9. Indeed, we additionally carried out the measured viscoelastic properties of CNC and SSY to compare the DNA's case, which can explain the non-Newtonian behavior of covalent bonded building blocks of DNA and CNC (Please see Supplementary Figure 4d).

Page 9

Similar microstructures made of cellulose nanocrystal (CNC) and sunset yellow (SSY) were investigated and were found to show almost identical and totally different morphologies, respectively (Supplementary Fig. 3). This can be explained by considering the aggregated shape of the molecules in the LC phase. In detail, DNA and CNC are made of chain-like long building blocks based on covalent bonds, while the SSY column is composed of very short discotic molecules stacked by noncovalent bonds (Supplementary Table 1).

In detail, lyotropic LC materials having the Col phase form a bend deformation parallel to the contact line during solvent evaporation. And then, two different phenomena can be observed as the contact line recedes, depending on the length of the building block. 1) The

short building block-based lyotropic LC materials, like SSY, form multiple domain walls due to undulations resulting from expansion and contraction near the interface, which look like cracks (Supplementary Fig. 3d-f)²⁶. 2) This undulation is suppressed in the chain-based lyotropic LC polymers as the concentration increases¹⁹, which is not observed in the highly concentrated condition³¹. This likely results in the differences among the three lyotropic LC materials during the formation of microstructures on the microposts, which cannot be easily explained by normal elastic energy under the steady state condition, as shown before³².

Page 10:

Accordingly, in this situation, the collective behaviour of the materials indicates that the chains can be stretched and/or aligned in some way due to capillary flow and/or contact line motion.

In addition, we performed dynamic viscosity measurements for the DNA, CNC, and SSY, and obtained quite different results. Shear thinning behaviour (reduced viscosity with increasing shear rate) was observed in the DNA and CNC solutions (Supplementary Fig. 4b, d), while the SSY solution showed Newtonian behaviour (Supplementary Fig. 4d). This can indirectly explain why the SSY solution did not produce stable microstructures in our platform (Supplementary Fig. 3d-f). A Newtonian fluid cannot produce collective behaviour due to the lack of elasticity effects. However, unfortunately, in this problem, it is extremely difficult to measure the exact elastic energy because their collective behaviour is motivated by the viscous drag effect, stretching the chains.

These varieties of bio- or biocompatible material-based microstructures should also be of interest and further investigated because each material exhibits unique characteristics, as well as hydrodynamic interaction, in the given condition.

Overall, my conclusion is that the article can be published after further cleaning of English and more rigorous use of physical terms.

> The article is edited by the native once again.

Reviewer #2 (Remarks to the Author):

The manuscript is much improved, especially from the initial submission. However, more description of the physical mechanism behind how $K_1 \gg K_2, K_3$ in your system would add to the quality of the paper, as mentioned in my last review (just a sentence or two). I also meant citation [19] by Smalyukh et. al. and not citation [21] in my previous review, but this is a minor mistake. I appreciate the authors looking more into the elastic energy argument, comparing the DNA system to CNC and SSY.

> Thanks for the kind comment. As recommended, the manuscript is revised like the below. In [Refs 28, 29], there are argues to deal with polymeric system, but to the best of our knowledge still the elastic energy in the columnar phase has not been exploited. So we used the elastic energy description in the nematic phase, which makes sense to understand the orientation of \mathbf{n}_{DNA} as shown in Figure 1.

Page 8-9:

In detail, these zigzags were the result of the balanced collective behaviour of DNA elasticity and shearing effect, as described in the following equation (1) of elastic free energy, which was derived in the N phase^{19,21},

$$f = \frac{K_1}{2} (\nabla \cdot \hat{n}) + \frac{K_2}{2} (\hat{n} \cdot (\nabla \times \hat{n}))^2 + \frac{K_3}{2} (\hat{n} \times (\nabla \times \hat{n}))^2 + \frac{E}{2} \frac{\partial \rho_p}{\rho_p^0} \square \square \quad (1)$$

where K_1 , K_2 and K_3 are the elastic constants for splay, twist and bend deformation, respectively, and $\partial \rho$ is the variation in the local DNA density, ρ^0 is the mean DNA density, and E is an elastic compressibility induced by dilative stress. In the given system, splay deformation is not usually found because $K_1 > K_2, K_3$ ^{29,30} at high concentration or low temperature or high molecular weight, so bend deformation is dominant, resulting in \mathbf{n}_{DNA} aligned parallel to the contact line (Fig.1b-e).

The language for the description of why CNC and your DNA system have similar behaviors, but not SSY, should be softened. In your revised manuscript, you write:

"This can be simply understood considering the aggregated shape in each case. In detail, DNA and CNC are made of chain-like building blocks, while SSY is composed of discotic molecule. This possibly changes the physical phenomena in terms of "stickiness" during the formation of microstructure on the microposts, which cannot be easily explained with the normal elastic energy under the steady state as

shown before."

It is better to make clear that this is a hypothesis, since you do not have direct experimental support of this. Since the elastic constant argument does not seem to explain the observed phenomena, the aggregated shape could be one possible explanation, but further experiments, outside the scope of the paper, are needed. Your chart detailing the common characteristics between the 3 materials that you included in the rebuttal should be included in supplementary materials and should be referenced in your description of why the elastic energy argument fails to describe what you observe. An additional few sentences detailing the elastic energy argument and why it fails is also needed.

> We appreciate your suggestion, the table is now added in supplementary materials (please see Supplementary Table 1). And the possible explanation is added in the highlight part in page 9 as we answered to reviewer 1's comment above.

The inclusion of Figure 5 in the text is a huge improvement to the manuscript. However, the color switching in Figure 5, c-f is not so obvious in the micrographs. Perhaps zooming in to the structure, instead of having a zoomed-out view, will help make this more evident.

> This is revised with the magnified images in Figure 5 like the below.

REVIEWERS' COMMENTS:

Reviewer #1 (Remarks to the Author):

I am now in favour of publication.

I hope that these and further experiments will foster a more detailed modelling and understanding of the rich observed behaviour.

Reviewer #2 (Remarks to the Author):

The authors have addressed my comments to my satisfaction.